# Comparison of Core Body Temperatures in Patients Administered Remimazolam or Propofol during Robotic-Assisted and Laparoscopic Radical Prostatectomy

**DOI:** 10.3390/medicina58050690

**Published:** 2022-05-23

**Authors:** Cheol Lee, Cheolhyeong Lee, Juhwan Lee, Gihyeon Jang, ByoungRyun Kim, SeongNam Park

**Affiliations:** 1Department of Anesthesiology and Pain Medicine, Wonkwang University School of Medicine Hospital, 895 Muwang-ro, Iksan 54538, Korea; leecheolhyeong@gmail.com (C.L.); fesjil@naver.com (J.L.); seaload0@naver.com (G.J.); 2Department of Obstetrics and Gynecology, Wonkwang University School of Medicine Hospital, 895 Muwang-ro, Iksan 54538, Korea; brkim74@wku.ac.kr

**Keywords:** anesthetic agents, thermoregulatory control, vasoconstriction threshold

## Abstract

*Background and Objectives:* Different types of anesthetics affect thermoregulatory mechanisms, such as the redistribution of body temperature, loss of skin heat, or inhibition of thermoregulatory vasoconstriction. Therefore, we compared remimazolam with propofol in terms of core body temperature in patients undergoing robotic-assisted and laparoscopic radical prostatectomy. *Materials and methods:* Ninety patients were randomly assigned to either the propofol–remifentanil (PR) group or the remimazolam–remifentanil (RR) group. The PR group (*n* = 45) received effect-site concentrations of 6.0 μg/mL of propofol and 4 ng/mL of remifentanil, followed by 0.9 mg/kg of 1% rocuronium and maintenance with effect-site concentrations of 2–4 μg/mL of propofol and 3 ng/mL of remifentanil. The RR group (*n* = 45) received remimazolam 6 mg/kg/h by continuous intravenous infusion and the effect-site concentration of 4 ng/mL of remifentanil, followed by 0.9 mg/kg of 1% rocuronium, remimazolam 1–3 mg/kg/h, and remifentanil 3 ng/mL. The primary outcome was core body temperature, and secondary outcomes included vasoconstriction threshold (°C) and time to onset of vasoconstriction (min). *Results:* The core body temperature in the RR group was significantly higher at 60, 80, 100, 120, 140, 160, and 180 min after induction than in the PR group (*p* < 0.01). The vasoconstriction threshold was significantly higher in the RR group (35.2 ± 0.4) than in the PR group (34.8 ± 0.3) (*p* < 0.01). The time to onset of vasoconstriction was significantly less in the RR group (150.5 ± 10.2) than in the PR group (158.5 ± 8.4) (*p* < 0.01). However, the incidence of intraoperative hypothermia was not significant between two groups. *Conclusions:* Remimazolam appears to reduce vasoconstriction threshold less than and had a faster onset of vasoconstriction, resulting in superior thermoregulatory control.

## 1. Introduction

The reduction in core temperature after anesthetic induction is mainly due to the inhibition of tonic thermoregulatory vasoconstriction by anesthetics and core to the peripheral redistribution of body heat [1]. The degree to which anesthetics lower the vasoconstriction threshold depends on the type and concentration of the drug. However, thermoregulation is also affected by non-thermoregulatory factors, including signals from the cardiovascular system [1,2,3].

Propofol has a strong vasodilatory effect in the central and peripheral regions, owing to its direct action on vascular smooth muscle and its blockade of the sympathetic nervous system. Due to this effect, the core to the peripheral redistribution of body heat is more pronounced. In addition, only the dose used for induction of anesthesia can exacerbate hypothermia [2,3,4].

The pharmacologic characteristics and clinical applications of remimazolam have been studied [5,6,7]. Remimazolam is a novel benzodiazepine that acts on the GABA-A receptor and has advantages such as a rapid onset, short duration of action, predictable and rapid recovery, hemodynamic stability, and mild respiratory depression [5]. At present, the literature is scarce on its effects on thermoregulation under general anesthesia.

We tested the hypothesis that the type of anesthetic could influence thermoregulatory mechanisms such as body temperature redistribution, skin heat loss, or inhibition of thermoregulatory vasoconstriction. Therefore, we compared remimazolam with propofol in terms of core body temperature, thermoregulatory vasoconstriction threshold, and time to onset of vasoconstriction in patients undergoing robotic-assisted radical prostatectomy (RARP) and laparoscopic radical prostatectomy (LRP). The vasoconstriction threshold is defined as the tympanic membrane temperature when the skin temperature gradient between the forearm and index finger temperature is 0 °C. The onset of vasoconstriction refers to the time to reach the vasoconstriction threshold.

## 2. Materials and Methods

### 2.1. Study Design and Patient Selection

This study was approved by the Institutional Review Board of Wonkwang University School of Medicine Hospital in South Korea (registration number 2021-02-08-009). The informed consent was obtained from all patients participating, and the trial was registered on https://clinicaltrials.gov/ (accessed on 1 April 2022) (NCT 05215834).

Ninety male patients aged between 19 and 75 years, with American Society of Anesthesiologists (ASA) physical status classes I–III, who were scheduled to undergo RARP or LRP, were included in this study. Patients taking medications that could affect cardiovascular function or body heat and those who had a body mass index of more than 30 kg/m^2^ or febrile illness were excluded.

### 2.2. Randomization

The randomization was stratified with a 1:1 allocation, using random block sizes of two. Participants were allocated (computerized random numbers) to the propofol–remifentanil (PR) group or the remimazolam–remifentanil (RR) group. Attending anesthesiologists who measured the outcomes were blinded to the study protocol.

### 2.3. Anesthesia Protocol

Before induction of anesthesia and in the post-anesthesia care unit, the patients’ core temperatures were measured by using an infrared tympanic membrane (TM) thermometer (ThermoScan IRT 4520, Braun, Melsungen, Germany), and the highest of three or more measurements was used with the thermometer on the patient’s right ear. The TM core body temperature was measured after removing earwax, using an otoscope.

After the introduction of general anesthesia, the nasopharyngeal temperature probe was inserted to a depth of 9.5 to 10 cm so that it was positioned at an appropriate location through the nostril. The nasopharyngeal core temperature was measured every 10 min until the end of surgery.

In the PR group (*n* = 45), the induction of anesthesia was commenced with an effect-site concentration of propofol 6.0 μg/mL and remifentanil 4 ng/mL until the loss of consciousness, followed by 1% rocuronium 0.9 mg/kg and maintained with an effect-site concentration of 2% propofol 4 μg/mL and remifentanil 3 ng/mL for a bispectral index (BIS) value of 40–60 until the end of surgery. In the RR group (*n* = 45), the induction of anesthesia was commenced with a continuous intravenous infusion of remimazolam 6 mg/kg/h and with the effect-site concentration of remifentanil 4 ng/mL until loss of consciousness, followed by 1% rocuronium 0.9 mg/kg, continuous intravenous infusion of remimazolam 1–3 mg/kg/h, and the effect-site concentration of remifentanil 3 ng/mL for a BIS value of 40–60 until the end of surgery.

Mechanical ventilation was commenced with a tidal volume of 8 mL/kg and a frequency of 12 breaths/min. A heat-and-moisture exchanger was used to heat and humidify the inhaled gases during anesthesia. Invasive arterial blood pressure, heart rate, electrocardiogram, oxygen saturation, and end-tidal concentrations of carbon dioxide were monitored. The temperature of the operating room was always maintained at 20–22 °C, and the corresponding relative humidity was 20–60%. Warmed intravenous or irrigating fluids and heated and humidified peritoneal insufflation gas (CO_2_) were administered during surgery.

Crystalloid (Ringer’s solution) and 6% HES 130/0.4 in a balanced electrolyte solution (Volulyte) were used concomitantly for fluid therapy in both groups. Transfusion of allogeneic red blood cells was administered for hemoglobin concentrations of ≤8.0 g/dL. An infusion fluid-heating apparatus (FMS2000, Belmont Instrument Corporation, Billerica, MA, USA)was used in both groups to warm the infused blood and fluid to 37 °C. A forced-air warming blanket (Bair Hugger Blanket, Augustine Medical, Inc., Eden Prairie, MN, USA) was used to deliver forced air at 38 °C and was applied to the upper body except for the surgical site.

In the present study, intraoperative hypothermia was defined as a case in which a core body temperature of a patient undergoing surgery under general anesthesia was below 36 °C. A mean arterial pressure (MAP) < 60 mmHg was considered hypotension. A heart rate (HR) < 50 beats/min was considered bradycardia.

### 2.4. Outcome Measures

The primary outcome was the comparison of the change of core body temperature in two drugs administered during general anesthesia. Secondary outcomes included vasoconstriction, time to onset of vasoconstriction, intraoperative hypothermia, mean arterial pressure, and heart rate during anesthesia.

### 2.5. Statistical Analysis

PASS 2008 (NCSS, Kaysville, UT, USA) software was used to calculate the sample size. A preliminary investigation had shown that the means ± standard deviations (SDs) of core body temperature at 60 min after induction as a primary outcome for the PR groups and the RR group were 35.64 ± 0.199 and 35.79 ± 0.243, respectively. The investigation revealed that a sample size of 36 patients per group would enable the detection of a significant difference with a power of 80% and an α-coefficient of 0.05. Allowing for a 20% dropout rate, the final sample size of this study was determined to be 45 patients per group. SPSS version 25.0 (SPSS Inc., Chicago, IL, USA) was used for the statistical analysis. Data are presented as the mean ± SD, median, or number (%) of patients. The groups were compared by using an independent t-test for continuous variables. Categorical variables were analyzed by using the chi-square test.

## 3. Results

A total of 120 patients were assessed for eligibility, and 30 were excluded because 17 did not meet the inclusion criteria, 10 refused to participate, and 3 had other reasons. Ninety patients were administered the study medication after randomization. Eleven patients were withdrawn due to conversion to open surgery and severe bleeding (Figure 1).

No significant differences between the groups were noted in age, body mass index, ASA class, type of surgery, duration of anesthesia, duration of surgery, amount of crystalloid and colloid administered, blood loss, transfusion, intraoperative hypothermia, hypotension, and bradycardia (Table 1).

The core body temperature in the RR group was significantly higher at 60, 80, 100, 120, 140, 160, and 180 min after induction than in the PR group (*p* < 0.01) (Figure 4). The mean arterial pressure and heart rate (Figure 5) were significantly higher in the RR group than in the PR group (*p* < 0.01).

## 4. Discussion

The main findings of our study demonstrated that the type of anesthetic agent might affect core body temperature through thermoregulatory mechanisms. Core body temperature in the RR group was significantly higher than in the PR group. The vasoconstriction threshold was significantly higher in the RR group than in the PR group. The time to onset of vasoconstriction was significantly shorter in the RR group than in the PR group.

Volatile anesthetics such as sevoflurane, the inhaled anesthetic nitric oxide, intravenous anesthetics such as propofol, and opioids used during anesthesia significantly impair thermoregulatory control. These drugs reduce the threshold for thermoregulatory vasoconstriction. The threshold reduction is dose-dependent and drug-dependent across the entire clinical spectrum [1,2,3,4]. The dose of remimazolam used in the present study showed earlier onset of vasoconstriction and a higher vasoconstriction threshold than those of propofol. As a result, the core body temperature with remimazolam use was higher than with propofol use from 60 min after induction.

Remimazolam and midazolam have the pharmacological profiles of classical benzodiazepines. The effect of midazolam on thermoregulatory vasoconstriction has also been investigated. A study reported that a dose of midazolam far exceeding those routinely used produced relatively little impairment of thermoregulatory control [8]. Another study reported that midazolam impaired tonic thermoregulatory vasoconstriction in a dose-dependent manner [9]. Reports on the effect of remimazolam on intraoperative thermoregulation are scarce. The dose of remimazolam used in the present study reduced the core body temperature by reducing the vasoconstriction threshold. The difference between the present study and the previous studies may result from the dosage of the anesthetics mentioned above. In the present study, an anesthetic dose was used, and in the previous studies, sedative doses were used.

With the usual combinations and doses of drugs used for general anesthesia, the thermoregulatory vasoconstriction threshold is reduced to approximately 34.5 °C. In the elderly, the vasoconstriction threshold decreases by approximately 1 °C [2]. In the present study, remifentanil was combined with propofol or remimazolam, and most patients were elderly. Vasoconstriction thresholds were 34.8 °C in the propofol-based PR group and 35.2 °C in the remimazolam-based RR group. These results may be due to passive or active warming techniques applied to the patients during surgery to prevent hypothermia.

Although the type and dose of intravenous anesthetics can affect thermoregulatory control, the remimazolam-based RR group, in terms of the incidence of intraoperative hypothermia, was not significant compared with propofol-base PR group in patients undergoing prolonged laparoscopic or robotic-assisted urologic major surgery in the present study. Prolonged laparoscopic procedures have been identified as a potential risk factor for hypothermia [10]. Further studies are needed in surgery of various durations to clarify the thermoregulatory control of remimazolam on core body temperature.

In the present study, remimazolam showed better outcomes in MAP and HR, but not in hypotension or bradycardia, compared with propofol. Previous studies have shown a lower incidence of hypotension in individuals administered remimazolam than in those administered propofol [5,6]. In the present study, remimazolam-based RR group required treatment for bradycardia or hypotension was not different compared to propofol-based PR group.

The vasodilatory properties of remimazolam and propofol may affect core body temperature. A previous study reported that the relationship between hemodynamic parameters and core body temperature was statistically significant but insufficient to attribute hemodynamic factors to core body temperature status [11]. In the present study, a decrease in MAP in both groups resulted in a decrease in core body temperature.

There are some limitations to the present study. First, the net effect of propofol and remimazolam could not be compared because of the use of remifentanil. Remifentanil inhibits the sympathetic nervous system and leads to vasodilation, which may affect thermoregulation, depending on the dose. The possibility that the effect of propofol or remimazolam was partially masked by remifentanil, thus reducing the measured differences, cannot be excluded [12]. Second, anesthetic-induced thermoregulatory inhibition is dose-dependent. Single doses of remimazolam and propofol were used in the present study to compare changes in core body temperature. Further clinical studies with various concentrations should be conducted to investigate the effect of remimazolam on core body temperature. Third, plasma catecholamine was not measured in the present study. A previous study [13] has reported the association between hypothermia and the catecholamine response in the perioperative period. Higher norepinephrine concentrations in the colder patients may represent a component of the thermoregulatory response. Fourth, the sample size of the present study was small, the age range was relatively narrow (66.4 ± 4.0 and 66.8 ± 4.0 years), and only male individuals were included. These factors may limit the generalizability of our results [14,15]. Therefore, additional studies should be conducted with larger sample sizes, greater age ranges, and both sexes to clarify the effect of remimazolam on core body temperature. Finally, we could not confirm that the mucosa in the upper or middle part of the nasopharynx was the optimal location for probe placement. A poorly positioned nasopharyngeal temperature probe yields core temperature values that are substantially different from those obtained with an optimally positioned probe [16].

## 5. Conclusions

In conclusion, remimazolam reduced the vasoconstriction threshold to a lesser extent than propofol and had an earlier onset of vasoconstriction, resulting in superior thermoregulatory control. In addition, remimazolam showed more stable hemodynamic parameters than propofol, including MAP and HR. Further studies are needed to clarify the relationship between autonomic thermoregulatory control and hemodynamic parameters in the use of remimazolam.

## Figures and Tables

**Figure 1 medicina-58-00690-f001:**
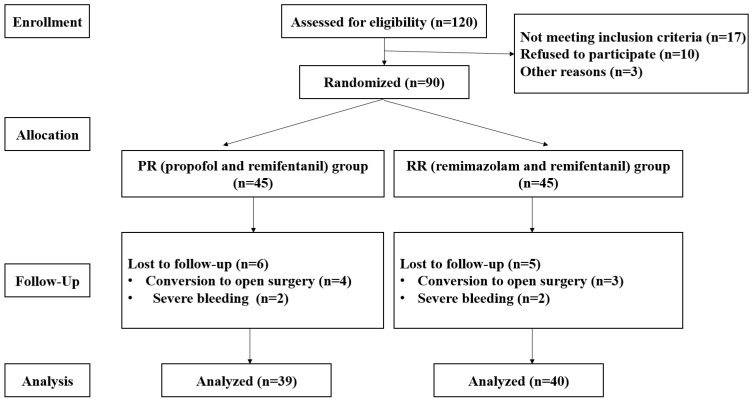
Consort flow diagram. The vasoconstriction threshold was significantly higher in the RR group (35.2 ± 0.4 °C) than in the PR group (34.8 ± 0.3 °C) (*p* < 0.01, 95% confidence interval (CI): −0.5 to −0.2). The time to onset of vasoconstriction was significantly shorter in the RR group (150.5 ± 10.2 min) than in the PR group (158.5 ± 8.4 min) (*p* < 0.01, 95% CI: 3.8–12.2) (Table 1 and Figure 2 and Figure 3).

**Figure 2 medicina-58-00690-f002:**
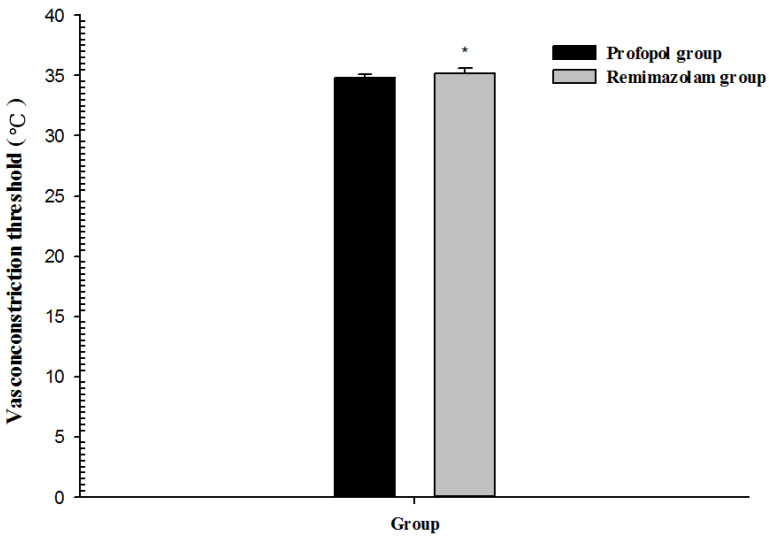
Vasoconstriction threshold after induction of general anesthesia with propofol or remimazolam; * *p* < 0.01 vs. propofol group.

**Figure 3 medicina-58-00690-f003:**
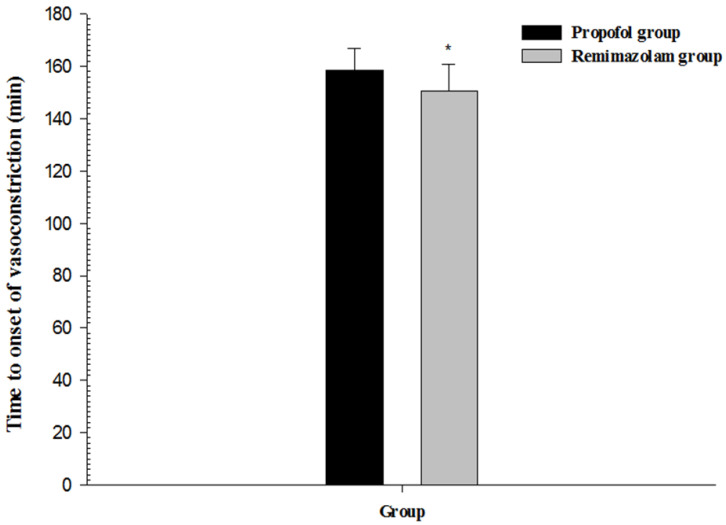
Time to onset of vasoconstriction after induction of general anesthesia with propofol or remimazolam; * *p* < 0.01 vs. propofol group.

**Figure 4 medicina-58-00690-f004:**
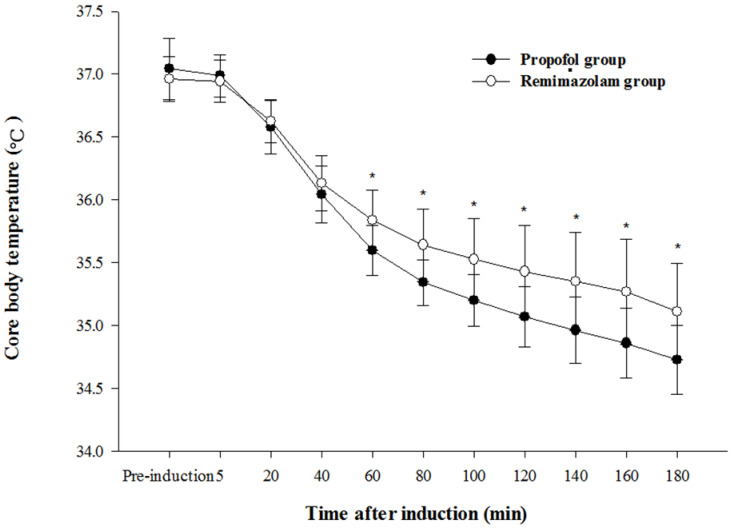
Core body temperature change after induction of general anesthesia with propofol or remimazolam; * *p* < 0.01 vs. propofol group.

**Figure 5 medicina-58-00690-f005:**
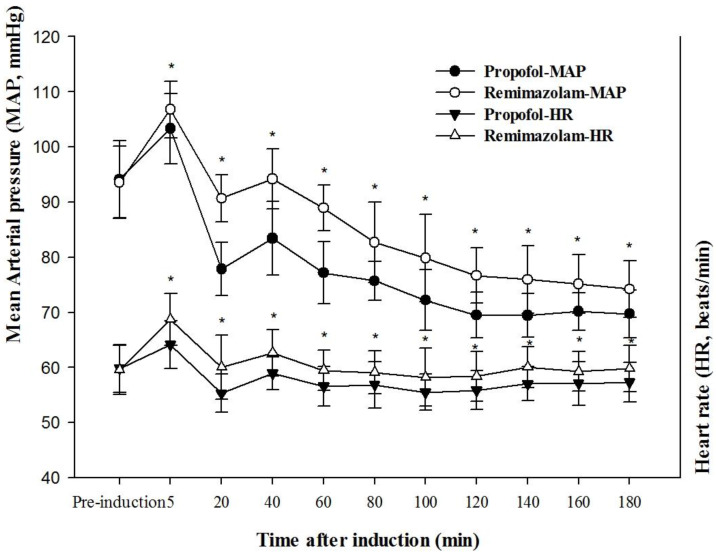
Mean arterial pressure and heart rate change after induction of general anesthesia with propofol or remimazolam; * *p* < 0.01 vs. propofol group.

**Table 1 medicina-58-00690-t001:** Demographic and intraoperative data.

	PR Group (*n* = 39)	RR Group (*n* = 40)	*p*-Value	95% CI
Age (years)	66.4 ± 4.0	66.8 ± 4.0	0.64	−2.2–1.4
Body mass index (kg/m^2^)	26.1 ± 1.4	26.2 ± 1.6	0.61	−0.8–0.5
ASA (I/II/III)	6/32/1	0/29/2	0.59	
Type of surgery			0.75	
LRP	17 (43.6)	16 (40)		
RARP	22 (56.4)	24 (60)		
Duration of anesthesia (min)	233.6 ± 14.7	231.9 ± 17.7	0.64	−5.6–9.0
Duration of surgery (min)	199.5 ± 14.2	196.9 ± 13.4	0.400	−3.6–8.8
Fluid type administered				
Crystalloid (mL)	1867.9 ± 162.4	1842.5 ± 150.4	0.47	−44.7–95.6
Colloid (mL)	315.4 ± 69.9	302.5 ± 66.0	0.40	−17.6–43.3
Blood loss (mL)	261.5 ± 70.2	252.5 ± 66.0	0.56	−21.5–39.6
Transfusion (allogenic red blood cell)	0 (0)	0 (0)		
Vasoconstriction threshold (°C)	34.8 ± 0.3	35.2 ± 0.4	<0.01	−0.5–−0.2
Time to onset of vasoconstriction (min)	158.5 ± 8.4	150.5 ± 10.2	<0.01	3.8–12.2
Intraoperative hypothermia	39 (100)	40 (100)		
Hypotension	0 (0)	0 (0)		
Bradycardia	0 (0)	0 (0)		

Values are expressed as mean ± standard deviation or number (%). CI, confidence interval; ASA, American Society of Anesthesiologists; LRP, laparoscopic radical prostatectomy; RARP, robot-assisted radical prostatectomy.

## Data Availability

Data can be made available upon reasonable request.

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
