# Peer review of "Comparison of Core Body Temperatures in Patients Administered Remimazolam or Propofol during Robotic-Assisted and Laparoscopic Radical Prostatectomy"

_medicina, 2022, doi:10.3390/medicina58050690_

Round 1

Reviewer 1 Report

I have reviewed an interesting paper by Cheol Lee at al. entitled “Comparison of core body temperatures in patients administered remimazolam or propofol during robotic-assisted and laparoscopic radical prostatectomy.” Here are my comments:

  1. Reference numbers should be placed in square brackets [ ], and placed before the punctuation; for example [1], [1–3] or [1,3]. For embedded citations in the text with pagination, use both parentheses and brackets to indicate the reference number and page numbers; for example [5] (p. 10). or [6] (pp. 101–105). (please see https://www.mdpi.com/journal/medicina/instructions) – discussion section should be revised accordingly
  2. Abstract section: “Conclusios: Remimazolam reduced vasoconstriction threshold less than and had a faster onset of vasoconstriction, resulting in superior thermoregulatory control.”

- correct the word “conclusion”

- there are some missing data here – please correct

- I think it will be better to say “Remimazolam appears to ……” - it is still a study with a limited number of patients to be able to give such a confident opinion

  1. Material and methods section:

- Was the temperature in the operating room measured? Is it comparable in both groups?

- Blood pressure is usually measured invasively for LRP? This is your standard protocol?

- How many anesthesiologists participated in the anesthesia of these patients?

- Who placed the nasopharyngeal temperature probe? (anesthesiologists, anesthesiology residents, nurse? How did you test that the position of nasopharyngeal temperature probes, placed blindly by anesthesia practitioners, was optimal? Various studies have shown that the positioning of probes is suboptimal in more than half of the cases.

- Which elements guided fluids administration?

- Were vasopressors used?

  1. If you have decided to have various sections in this article (introduction, ….), the last one should be named Discussion and conclusion, or you should have a new one for Conclusion section.

Author Response

We would like to thank the reviewers for their thoughtful responses, which we feel have improved the manuscript. Below are all of their suggestions and concerns followed by our responses

I have reviewed an interesting paper by Cheol Lee at al. entitled “Comparison of core body temperatures in patients administered remimazolam or propofol during robotic-assisted and laparoscopic radical prostatectomy.” Here are my comments:

  1. Reference numbers should be placed in square brackets [ ], and placed before the punctuation; for example [1], [1–3] or [1,3]. For embedded citations in the text with pagination, use both parentheses and brackets to indicate the reference number and page numbers; for example [5] (p. 10). or [6] (pp. 101–105). (please see https://www.mdpi.com/journal/medicina/instructions) – discussion section should be revised accordingly

Answer:  Agreed, We have revised the reference numbers in the discussion

  1. Abstract section: “Conclusios: Remimazolam reduced vasoconstriction threshold less than and had a faster onset of vasoconstriction, resulting in superior thermoregulatory control.”

- correct the word “conclusion”

Answer:   Agreed, We have changed “conclusios” to “conclusions”

- there are some missing data here – please correct

Answer:   We have added the texts in Abstract and Discussion and conclusions

Abstract:

However, the incidence of intraoperative hypothermia was not significant between two

groups.

Discussion and conclusions

Although the type and dose of intravenous anesthetics can affect thermoregulatory control, remizolam-base RR group, in terms of the incidence of intraoperative hypothermia was not significant compared with propofol-base PR group in patients undergoing prolonged laparoscopic or robotic-assisted urologic major surgery in the present study. Prolonged laparoscopic procedures have been identified as a potential risk factor for hypothermia [10]. Further studies are needed in surgery of various durations to clarify the thermoregulatory control of remimazolam on core body temperature.

- I think it will be better to say “Remimazolam appears to ……” - it is still a study with a limited number of patients to be able to give such a confident opinion

Answer:  Agreed, We have modified the phrase, as you recommended.

Remimazolam appears to reduce the vasoconstriction threshold

  1. Material and methods section:

- Was the temperature in the operating room measured? Is it comparable in both groups?

Answer:  Before induction of anesthesia in the operating room and the postanesthesia care unit, the patients’ core temperatures in both groups were measured

- Blood pressure is usually measured invasively for LRP? This is your standard protocol?

Answer: A heat and moisture exchanger was used to heat and humidify the inhaled gases during anesthesia. Invasive arterial blood pressure, heart rate, electrocardiogram, oxygen saturation, and end-tidal concentrations of carbon dioxide were monitored.

These monitoring are routine in major surgery in our hospital.

- How many anesthesiologists participated in the anesthesia of these patients?

Answer:   In our hospital, an attending anesthesiologist and a resident doctor take part in anesthesia for major surgery

- Who placed the nasopharyngeal temperature probe? (anesthesiologists, anesthesiology residents, nurses? How did you test that the position of nasopharyngeal temperature probes, placed blindly by anesthesia practitioners, was optimal? Various studies have shown that the positioning of probes is suboptimal in more than half of the cases.

Answer:  Agreed, An attending anesthesiologist placed the nasopharyngeal temperature probe. The use of bronchoscoy is advised by an anesthesiologist to confirm proper placement. We couldn’t use bronchoscopy to confirm the optimal position of the nasopharyngeal temperature probe in this study. Therefore, we mentioned it in the limitation section of the discussion.

- Which elements guided fluids administration?

Answer: Crystalloid (Ringer’s solution) and 6% HES 130/0.4 in a balanced electrolyte solution are routinely used in our hospital.

Three major indications exist for intravenous fluid administration.

  1. Resuscitation fluids are used to correct an intravascular volume deficit or acute hypovolemia.
  2. Replacement solutions are prescribed to correct existing or developing deficits that cannot be compensated by oral intake alone.
  3. Maintenance solutions are indicated in hemodynamically stable patients that are not able/allowed to drink water to cover their daily requirements of water and electrolytes

- Were vasopressors used?

Answer:   We didn’t use a vasopressor, because it can affect the thermoregulatory vasoconstriction threshold.

  1. If you have decided to have various sections in this article (introduction, ….), the last one should be named Discussion and conclusion, or you should have a new one for the Conclusion section.

Answer:  Agreed, We have changed “Discussion” to “Discussion and conclusions”.

Reviewer 2 Report

Well conducted and presented study. 

Specific comments: 

The thermal vasoconstriction index should be explained when it is mentioned first

How was onset of vasoconstriction measured, please explain briefly

Figure 3, y-axis: Time to ...

Author Response

Specific comments: 

The thermal vasoconstriction index should be explained when it is mentioned first

Answer: Agreed, we have explained the thermal vasoconstriction index in the introduction

“The vasoconstriction threshold is defined as the tympanic membrane temperature when the skin temperature gradient between the forearm and index finger temperature is 0°C. The onset of vasoconstriction refers to the time to reach the vasoconstriction threshold”.

How was onset of vasoconstriction measured, please explain briefly

Answer: As described above, We have explained how to measure the onset of vasoconstriction.

The vasoconstriction threshold is defined as the tympanic membrane temperature when the skin temperature gradient between the forearm and index finger temperature is 0°C. The onset of vasoconstriction refers to the time to reach the vasoconstriction threshold.

Figure 3, y-axis: Time to

Answer: Agreed, We have changed “Times” to “Time” in Figure 3, as you recommended